# TK-KNN: A Balanced Distance-Based Pseudo Labeling Approach for Semi-Supervised Intent Classification

**Nicholas Botzer♣, David Vasquez♠, Tim Weninger♣, Issam Laradji♠**
♣University of Notre Dame, ♠ServiceNow Research
♣nbotzer@nd.edu, ♠issam.laradji@gmail.com

## Abstract

The ability to detect intent in dialogue systems has become increasingly important in modern technology. These systems often generate a large amount of unlabeled data, and manually labeling this data requires substantial human effort. Semi-supervised methods attempt to remedy this cost by using a model trained on a few labeled examples and then by assigning pseudo-labels to further a subset of unlabeled examples that has a model prediction confidence higher than a certain threshold. However, one particularly perilous consequence of these methods is the risk of picking an imbalanced set of examples across classes, which could lead to poor labels. In the present work, we describe Top-K K-Nearest Neighbor (TK-KNN), which uses a more robust pseudo-labeling approach based on distance in the embedding space while maintaining a balanced set of pseudo-labeled examples across classes through a ranking-based approach. Experiments on several datasets show that TK-KNN outperforms existing models, particularly when labeled data is scarce on popular datasets such as CLINC150 and Banking77. Code is available at https://github.com/ServiceNow/tk-knn

## 1 Introduction.

Large language models like BERT [Devlin et al., 2018] have significantly pushed the boundaries of Natural Language Understanding (NLU) and created interesting applications such as automatic ticket resolution [Marcuzzo et al., 2022]. A key component of such systems is a virtual agent's ability to understand a user's intent to respond appropriately. Successful implementation and deployment of models for these systems require a large amount of labeled data to be effective. Although deployment of these systems often generate a large amount of data that could be used for fine-tuning, the cost of labeling this data is high. Semi-supervised learning methodologies are an ob-

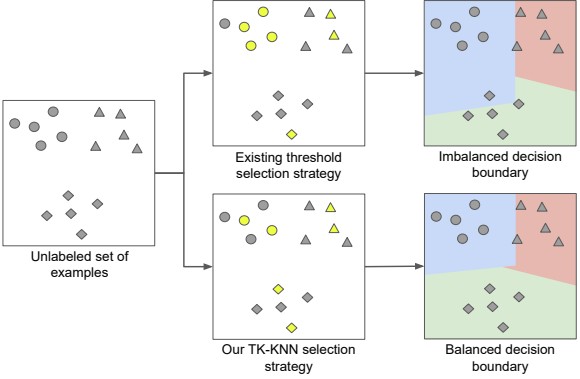

Figure 1: Example of pseudo label selection when using a threshold (top) versus the top-k sampling strategy (bottom). In this toy scenario, we chose $k = 2$, where each class is represented by a unique shape. As the threshold selection strategy pseudo-labels data elements (shown as yellow) that exceed the confidence level, the model tends to become biased towards classes that are easier to predict. This bias causes a cascade of mislabels that leads to even more bias towards the majority class.

vious solution because they can significantly reduce the amount of human effort required to train these kinds of models [Laradji et al., 2021, Zhu and Goldberg, 2009] especially in image classification tasks [Zhai et al., 2019, Ouali et al., 2020b, Yalniz et al., 2019]. However, as well shall see, applications of these models is difficult for NLU and intent classification because of the label distribution.

Indeed, research most closely realted to the present work is the Slot-List model by Basu et al. [2021], which focuses on the meta-learning aspect of semi-supervised learning rather than using unlabeled data. In a similar vein the GAN-BERT [Croce et al., 2020] model shows that using an adversarial learning regime can be devised to ensure that the extracted BERT features are similar amongst the unlabeled and the labeled data sets and substantially boost classification performance. Other methods have investigated how data augmen-

tation can be applied to the NLP domain to enforce consistency in the models [Chen et al., 2020], and several other methods have been proposed from the computer vision community. However, a recent empirical study found that many of these methods do not provide the same benefit to NLU tasks as they provide to computer vision tasks [Chen et al., 2021] and can even hinder performance.

Intent classification remains a challenging problem for multiple reasons. Generally, the number of intents a system must consider is relatively large, with sixty classes or more. On top of that, most queries consists of only a short sentence or two. This forces models to need many examples in order to learn nuance between different intents within the same domain. In the semi-supervised setting, many methods set a confidence threshold for the model and assign pseudo-labels to the unlabeled data if their confidence is above the threshold [Sohn et al., 2020]. This strategy permits high-confidence pseudo-labeled data elements to be included in the training set, which typically results in performance gains. Unfortunately, this approach also causes the model to become overconfident for classes that are easier to predict.

In the present work, we describe the Top-K K-Nearest Neighbor (TK-KNN) method for training semi-supervised models. The main idea of this method is illustrated in Figure 1. TK-KNN makes two improvements over other pseudo-labeling approaches. First, to address the model overconfidence problem, we use a top-k sampling strategy when assigning pseudo-labels to enforce a balanced set of classes by taking the top-k predictions per class, not simply the predictions that exceed a confidence threshold overall predictions. Furthermore, when selecting the top-k examples the sampling strategy does not simply rely on the model's predictions, which tend to be noisy. Instead we leverage the embedding space of the labeled and unlabeled examples to find those with similar embeddings and combine them with the models' predictions. Experiments using standard performance metrics of intent classification are performed on three datasets: CLINC150 [Larson et al., 2019], Banking77 [Casanueva et al., 2020], and Hwu64 [Liu et al., 2019]. We find that the TK-KNN method outperforms existing methods in all scenarios and performs exceptionally well in the low-data scenarios.

## 2 Related Work

**Intent Classification** The task of intent classification has attracted much attention in recent years due to the increasing use of virtual customer service agents. Recent research into intent classification systems has mainly focused on learning out of distribution data [Zhan et al., 2021, Zhang et al., 2021b, Cheng et al., 2022, Zhou et al., 2022]. These techniques configure their experiments to learn from a reduced number of the classes and treat the remaining classes as out-of-distribution during testing. Although this research is indeed important in its own regard, it deviates from the present work's focus on semi-supervised learning.

**Pseudo Labeling** Pseudo labeling is a mainstay in semi-supervised learning [Lee et al., 2013, Rizve et al., 2021, Cascante-Bonilla et al., 2021]. In simple terms, pseudo labelling uses the model itself to acquire hard labels for each of the unlabeled data elements. This is achieved by taking the argmax of the models' output and treating the resulting label as the example's label. In this learning regime, the hard labels are assigned to the unlabeled examples without considering the confidence of the model's predictions. These pseudo-labeled examples are then combined with the labeled data to train the model iteratively. The model is then expected to iteratively improve until convergence. The main drawback of this method is that mislabeled data elements early in training can severely degrade the performance of the system.

A common practice to help alleviate mislabeled samples is to use a threshold $\tau$ to ensure that only high-quality (*i.e.*, confident) labels are retained [Sohn et al., 2020]. The addition of confidence restrictions into the training process [Sohn et al., 2020] has shown improvements but also restricts the data used at inference time and introduces the confidence threshold value as yet another hyperparameter that needs to be tuned.

Another major drawback of this selection method is that the model can become very biased towards the easy classes in the early iterations of learning [Arazo et al., 2020]. Recent methods, such as FlexMatch [Zhang et al., 2021a], have discussed this problem at length and attempted to address this issue with a curriculum learning paradigm that allows each class to have its own threshold. These thresholds tend to be higher for majority classes lower for less-common classes. However, this only

serves to exacerbate the problem because the less-common classes will have less-confident labels. A previous work by Zou et al. [2018] proposes a similar class balancing parameter to be learned per class, but is applied to the task of unsupervised domain adaptation. A close previous work to ours is co-training [Nigam and Ghani, 2000] that iteratively adds a single example from each class throughout the self-training. Another more recent work [Gera et al., 2022] also proposes a balanced sampling mechanism for self-training, but starts from a zero-shot perspective and limits to two cycles of self-training.

Another pertinent work is by Chen et al. [2022], who introduce ContrastNet, a framework that leverages contrastive learning for few-shot text classification. This is particularly relevant to our study considering the challenges posed by datasets with a scarce number of labeled examples per class. A notable work by Wang et al. [2022] employs a contrastive learning-enhanced nearest neighbor mechanism for multi-label text classification, which bears some resemblance to the KNN strategies discussed in our work.

The TK-KNN strategy described in the present work addresses these issues by learning the decision boundaries for all classes in a balanced way while still giving preference to accurate labels by considering the proximity between the labeled and the unlabeled examples in the embedding space.

**Distance-based Pseudo labeling** Another direction explored in recent work is to consider the smoothness and clustering assumptions found in semi-supervised learning [Ouali et al., 2020a] for pseudo labeling. The smoothness assumption states that if two points lie in a high-density region their outputs should be the same. The clustering assumption similarly states that if points are in the same cluster, they are likely from the same class. Recent work by Zhu et al. [2022] propose a training-free approach to detect corrupted labels. They use a k-style approach to detect corrupted labels that share similar features. The results of this work show that the smoothness and clustering assumptions are also applicable in a latent embedding space and therefore data elements that are close in the latent space are likely to share the same clean label.

Two other recent works have made use of these assumptions in semi-supervised learning to improve their pseudo-labeling process. First, Taherkhani et al. [2021] use the Wasserstein dis-

tance to match clusters of unlabeled examples to labeled clusters for pseudo-labeling.

Second, the aptly-named feature affinity based pseudo-labeling [Ding et al., 2019] method uses the cosine similarity between unlabeled examples and cluster centers that have been discovered for each class. The selected pseudo label is determined based on the highest similarity score calculated for the unlabeled example.

Results from both of these works demonstrate that distance-based pseudo-labeling strategies yield significant improvements over previous methods. However, both of these methods depend on clusters formed from the labeled data. In the intent classification task considered in the current study, the datasets sometimes have an extremely limited number of labeled examples per class, with instances where there is only one labeled example per class. This scarcity of labeled data makes forming reliable clusters quite challenging. Therefore, the TK-KNN model described in the present work adapted the K-Nearest Neighbors search strategy to help guide our pseudo-labeling process.

## 3 Top-K KNN Semi-Supervised Learning

### 3.1 Problem Definition

We formulate the problem of semi-supervised intent classification as follows:

Given a set of labeled intents $X = (x_n, y_n) : n \in (1, ..., N)$ where $x_n$ represents the intent example and $y_n$ the corresponding intent class $c \in C$ and a set of unlabeled intents $U = u_m : m \in (1, ..., M)$, where each instance $u_m$ is an intent example lacking a label. Intents are fed to the model as input and the model outputs a predicted intent class, denoted as $p_{model}(c|x, \theta)$, where $\theta$ represents some pre-trained model parameters. Our goal is to learn the optimal parameters for $\theta$.

### 3.2 Method Overview

As described above, we first employ pseudo labeling to iteratively train (and re-train) a model based on its most confident-past predictions. In the first training cycle, the model is trained on only the small portion of labeled data $X$. In the subsequent cycles, the model is trained on the union of $X$ and a subset of the unlabeled data $U$ that has been pseudo-labeled by the model in the previous cycle. Figure 2 illustrates an example of this training regime with the TK-KNN method.

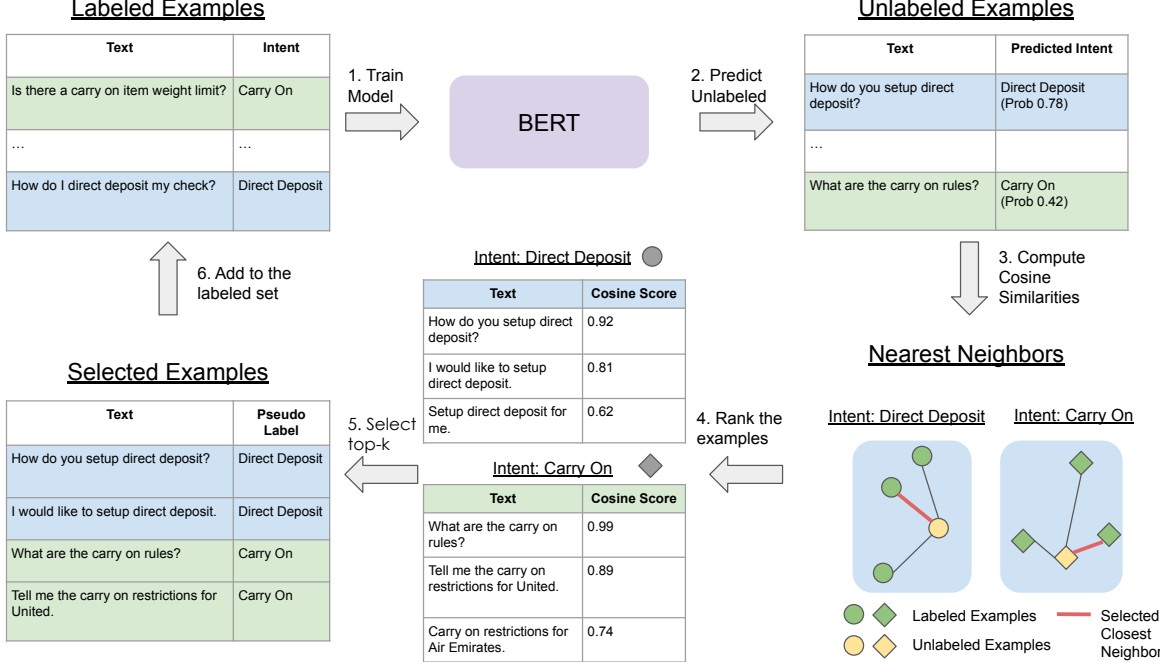

Figure 2: TK-KNN overview. The model is (1) trained on the small portion of labeled data. Then, this model is used to predict (2) pseudo labels on the unlabeled data. Then the cosine similarity (3) is calculated for each unlabeled data point with respect to the labeled data points in each class. Yellow shapes represent unlabeled data and green represent labeled data. Similarities are computed and unlabeled examples are ranked (4) based on a combination of their predicted probabilities and cosine similarities. Then, the top-k ($k = 2$) examples are selected (5) for each class. These examples are finally added (6) to the labeled dataset to continue the iterative learning process.

We use the BERT-base [Devlin et al., 2018] model with an added classification head to the top. The classification head consists of a dropout layer followed by a linear layer with dropout and ends with an output layer that represents the dataset's class set $C$. We select the BERT-base model for fair comparison with other methods.

## 3.3 Top-K Sampling

When applying pseudo-labeling, it is often observed that some classes are easier to predict than others. In practice, this causes the model to become biased towards the easier classes [Arazo et al., 2020] and perform poorly on the more difficult ones. The Top-K sampling process within the TK-KNN system seeks to alleviate this issue by growing the pseudo-label set across all labels together.

When we perform pseudo labeling, we select the **top-k** predictions per class from the unlabeled data. This selection neither uses nor requires any threshold; instead, it limits each class to choose the predictions with the highest confidence. We rank each predicted data element with a score based on the models predicted probability.

$$\text{score}(u_m) = p_{model}(y = c | u_m; \theta) \qquad (1)$$

After each training cycle, the number of pseudo labels in the dataset will have increased by $k$ times the number of classes. This process continues until all examples are labeled or some number of pre-defined cycles has been reached. We employ standard early stopping criteria [Prechelt, 1998] during each training cycle to determine whether or not to stop training.

## 3.4 KNN-Alignment

Although our top-k selection strategy helps alleviate the model's bias, it still relies entirely on the model predictions. To enhance our top-k selection strategy, we utilize a KNN search to modify the scoring function that is used to rank which pseudo-labeled examples should be included in the next training iteration. The intuition for the use of the KNN search comes from the findings in [Zhu et al., 2022] where "closer" instances are more likely to share the same label based on the neighborhood information when some labels are corrupted, which often occurs in semi-supervised learning from the pseudo-labeling strategy.

Specifically, we extract a latent representation from each example in our training dataset, both the labeled and unlabeled examples. We formulate this latent representation in the same way as Sentence-BERT [Reimers and Gurevych, 2019] to construct a robust sentence representation. This representation is defined as the mean-pooled representation of the final BERT layer that we formally define as:

$$z = \mathsf{mean}([CLS], T_1, T_2, ..., T_M) \qquad (2)$$

Where CLS is the class token, $T$ is each token in the sequence, $M$ is the sequence length, and $z$ is the extracted latent representation. When we perform our pseudo labeling process we extract the latent representation for all of our labeled data $X$ as well as our unlabeled data $U$.

For each unlabeled example, we calculate the cosine similarity between its latent representation and the latent representations of the labeled counterparts belonging to the predicted class.

The highest cosine similarity score between the unlabeled example and its labeled neighbors is used to calculate the score of an unlabeled example. Let $z_m$ and $z_n$ be the latent representations of the unlabeled data point $u_m$ and a labeled data point $x_n$, respectively. An additional hyperparameter, $\beta$, permits the weighing of the model's prediction and the cosine similarity for the final scoring function.

$$\begin{aligned} \mathrm{score}(u_m) = (1 - \beta) \times p_{model}(y|u_m; \theta) + \\ \beta \times \mathrm{sim}(z_n, z_m) \end{aligned} \qquad (3)$$

In this equation $\mathrm{sim}(z_n, z_m)$ computes the cosine similarity between the latent representations of $u_m$ and its closest labeled counterpart in the predicted class. With these scores we then follow the previously discussed top-k selection strategy to ensure balanced classes. The addition of the K-nearest neighbor search helps us to select more accurate labels early in the learning process. We provide pseudo code for our pseudo-labeling strategy in Appendix Algorithm 1.

### 3.5 Loss Function

As we use the cosine similarity to help our ranking method we want to ensure that similar examples are grouped together in the latent space. While the cross entropy loss is an ideal choice for classification, as it incentivizes the model to produce accurate predictions, it does not guarantee that discriminative features will be learned [Elsayed et al., 2018],

which our pseudo labeling relies on. To address this issue, we supplemented the cross-entropy loss with a supervised contrastive loss [Khosla et al., 2020] and a differential entropy regularization loss [Sablayrolles et al., 2019], and trained the model using all three losses jointly.

$$L_{CE} = -\sum_{i=1}^{C} y_i \log(\hat{y}_i) \qquad (4)$$

We select the supervised contrastive loss as shown in [Khosla et al., 2020]. This ensures that our model with learn good discriminative features in the latent space that separate examples belonging to different classes. The supervised contrastive loss relies on augmentations of the original examples. To get this augmentation we simply apply a dropout layer of $0.2$ to the representations that we extract from the model. As is standard for the supervised contrastive loss we add a separate projection layer to our model to align the representations. The representations fed to the projection layer is the mean-pooled BERT representation as shown in Eq. 2.

$$L_{SCL} = \sum_{i \in I} \frac{-1}{|P(i)|} \sum_{p \in P(i)} log \frac{sim(z_i, z_p)/\tau}{\sum_{a \in A(i)} sim(z_i, z_a)/\tau} \qquad (5)$$

When adopting the contrastive loss previous works [El-Nouby et al., 2021] have discussed how the model can collapse in dimensions as a result of the loss. We follow this work in adopting a differential entropy regularizer in order to spread the representations our more uniformly. The method we use is based on the Kozachenko and Leonenko [1987] differential entropy estimator:

$$L_{KoLeo} = -\frac{1}{N} \sum_{i=1}^{N} \log(p_i) \qquad (6)$$

Where $p_i = min_{(i \neq j)} ||f(x_i) - f(x_j)||$. This regularization helps to maximize the distance between each point and its neighbors. By doing so it helps to alleviate the rank collapse issue. We combine this term with the cross-entropy and contrastive objectives, weighting it using a coefficient $\gamma$.

$$L_{ALL} = L_{CE} + L_{SCL} + \gamma L_{KoLeo} \qquad (7)$$

The joint training of these individual components leads our model to have better discriminative

features that are more robust, that results in improved generalization and prediction accuracy.

## 4 Experiments

### 4.1 Experimental Settings

**Datasets** We use three well-known benchmark datasets to test and compare the TK-KNN model against other models on the intent classification task. Our intent classification datasets are **CLINC150** [Larson et al., 2019] that contains 150 in-domain intents classes from ten different domains and one out-of-domain class. **BANKING77** [Casanueva et al., 2020] that contains 77 intents, all related to the banking domain. **HWU64** [Liu et al., 2019] which includes 64 intents coming from 21 different domains. Banking77 and Hwu64 do not provide validation sets, so we created our own from the original training sets. All datasets are in English. A breakdown of each dataset is shown in Table 1.

| Dataset | Intents | Domain | Train | Val | Test |
|---------|---------|--------|-------|-----|------|
| CLINC150 | 151 | 10 | 15,250 | 3,100 | 5,550 |
| Banking77 | 77 | 1 | 9,002 | 1,001 | 3,080 |
| Hwu64 | 64 | 21 | 8,884 | 1,076 | 1,076 |

Table 1: Breakdown of the intent classification datasets. Note that BANKING77 and HWU64 do not provide validation sets, so we generated a validation set from the original training set.

We conducted our experiments with varying amounts of labeled data for each dataset. All methods are run with five random seeds and the mean average accuracy of their results are reported along with their 95% confidence intervals [Dror et al., 2018].

### 4.2 Baselines

To perform a proper and thorough comparison of TK-KNN with existing methods, we implemented and repeated the experiments on the following models and strategies.

- **Supervised**: Use only labeled portion of dataset to train the model without any semi-supervised training. This model constitutes a competitive lower bound of performance because of the limits in the amount of labeled data.

- **Pseudo Labeling (PL)** [Lee et al., 2013]: This strategy trains the model to convergence then makes predictions on all of the unlabeled

data examples. These examples are then combined with the labeled data and used to re-train the model in an iterative manner.

- **Pseudo Labeling with Threshold (PL-T)** [Sohn et al., 2020]: This process follows the pseudo labeling strategy but only selects unlabeled data elements which are predicted above a threshold $\tau$. We use a $\tau$ of 0.95 based on the findings from previous work.

- **Pseudo Labeling with Flexmatch (PL-Flex)** [Zhang et al., 2021a]: Rather than using a static threshold across all classes, a dynamic threshold is used for each class based on a curriculum learning framework.

- **GAN-BERT** [Croce et al., 2020]: This method applies generative adversarial networks [Goodfellow et al., 2020] to a pretrained BERT model. The generator is an MLP that takes in a noise vector. The output head added to the BERT model acts as the discriminator and includes an extra class for predicting whether a given data element is real or not.

- **MixText** [Chen et al., 2020]: This method extends the MixUp [Zhang et al., 2017] framework to NLP and uses the hidden representation of BERT to mix together. The method also takes advantage of consistency regularization in the form of back translated examples.

- **TK-KNN** : The method described in the present work using top-k sampling with a weighted selection based on model predictions and cosine similarity to the labeled samples.

### 4.3 Implementation Details

Each method uses the BERT base model with a classification head attached. We use the base BERT implementation provided by Huggingface Wolf et al., 2019, that contains a total of 110M parameters. All models are trained for 30 cycles of self-training. The models are optimized with the AdamW optimizer with a learning rate of 5e-5. Each model is trained until convergence by early stopping applied according to the validation set. We use a batch size of 256 across experiments and limit the sequence length to 64 tokens. For TK-KNN, we set $k = 6$, $\beta = 0.75$, and $\gamma = 0.1$ and report the results for

| Method | Percent Labeled | | | |
|---|---|---|---|---|
| | 1% | 2% | 5% | 10% |
| CLINC150 | | | | |
| Supervised | 27.35 ±1.71 | 49.15 ±1.99 | 67.96 ±0.85 | 75.05 ±1.57 |
| PL | 24.51 ±3.92 | 48.58 ±1.79 | 69.19 ±0.54 | 76.92 ±1.05 |
| PL-T | 39.05 ±3.26 | 56.65 ±1.53 | 71.25 ±0.5 | 79.29 ±1.62 |
| PL-Flex | 42.81 ±4.39 | 60.07 ±1.42 | 73.42 ±1.62 | 78.86 ±1.01 |
| GAN-BERT | 18.18 ±0.0 | 23.29 ±11.42 | 44.89 ±24.39 | 63.02 ±25.1 |
| MixText | 12.86 ±6.39 | 37.93 ±16.8 | 61.39 ±0.77 | 74.29 ±0.37 |
| TK-KNN | **53.73** ±1.72 | **65.87** ±1.18 | **74.31** ±0.96 | **79.45** ±1.01 |
| BANKING77 | | | | |
| Supervised | 34.73 ±1.5 | 47.51 ±2.89 | 70.27 ±1.08 | 80.82 ±0.41 |
| PL | 29.09 ±3.83 | 45.16 ±2.71 | 69.69 ±2.16 | 80.26 ±0.49 |
| PL-T | 35.12 ±3.86 | 51.67 ±3.14 | 71.16 ±1.98 | 81.88 ±0.43 |
| PL-Flex | 40.04 ±3.4 | 54.18 ±3.31 | 73.43 ±1.55 | 82.54 ±0.84 |
| GAN-BERT | 5.4 ±9.16 | 16.98 ±21.73 | 54.09 ±29.56 | 79.64 ±1.39 |
| MixText | 32.73 ±6.02 | 54.75 ±3.15 | 76.59 ±1.05 | 82.34 ±0.94 |
| TK-KNN | **54.16** ±4.56 | **62.71** ±2.30 | **76.73** ±01.46 | **84.45** ±0.52 |
| HWU64 | | | | |
| Supervised | 48.87 ±1.55 | 63.88 ±1.6 | 74.67 ±1.91 | 82.21 ±1.72 |
| PL | 48.46 ±1.86 | 64.39 ±1.66 | 75.76 ±1.69 | 82.49 ±0.94 |
| PL-T | 56.9 ±1.64 | 68.29 ±1.79 | 76.9 ±1.1 | 82.96 ±1.69 |
| PL-Flex | 60.15 ±3.27 | 69.87 ±0.93 | 77.99 ±1.4 | 83.83 ±1.2 |
| GAN-BERT | 33.36 ±16.55 | 32.9 ±29.07 | 72.32 ±1.41 | 81.78 ±1.64 |
| MixText | 33.3 ±8.98 | 56.46 ±11.08 | 66.65 ±7.28 | 79.72 ±1.27 |
| TK-KNN | **65.33** ±2.29 | **73.03** ±1.31 | **79.63** ±0.56 | **84.59** ±0.58 |

Table 2: Mean test accuracy results and their 95% confidence intervals across 5 repetitions with different different random seeds. All experiments used $k = 6$ and $\beta = 0.75$. TK-KNN outperformed existing state of the art models, especially when the label set is small. The confidence interval also shows that the TK-KNN results were stable across repetitions.

these settings. An ablation study of these two hyperparameters is presented later. For details on the settings used for MixText please see Appendix B.

**Computational Use.** In total we estimate that we used around 18,000 GPU hours for this project. For the final experiments and ablation studies we estimate that the TK-KNN model used 4400 GPU hours. Experiments were carried out on Nvidia Tesla P100 GPUs that each had 12GB of memory and 16GB of memory.

## 5 Results

Results from these experiments are shown in Table 2. These quantitative results demonstrate that TK-KNN yielded the best performance on the benchmark datasets. We observed the most significant performance gains for CLINC150 and BANKING77, where these datasets have more classes. For instance, on the CLINC150 dataset with 1% labeled data, our method performs **10.92%** better than the second best strategy, FlexMatch. As the portion of labeled data used increases, we notice that the effectiveness of TK-KNN diminishes.

Another observation from these results is that the GAN-BERT model tends to be unstable when the

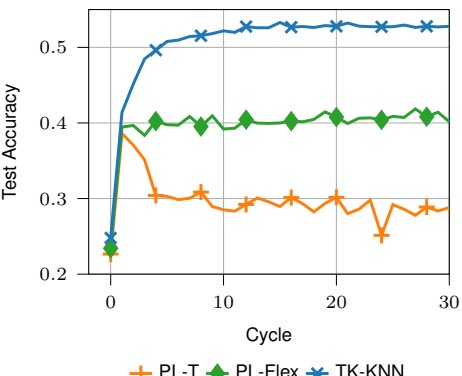

Figure 3: Convergence analysis of pseudo-labelling strategies on CLINC150 at 1% labeled data. TK-KNN clearly outperforms the other pseudo-labelling strategies by balancing class pseudo labels after each training cycle.

labeled data is limited. However, GAN-BERT does improve as the proportion of labeled data increases. This is consistent with previous findings on training GANs from computer vision [Salimans et al., 2016, Arjovsky and Bottou, 2017]. We also find that while the MixText method shows improvements the benefits of consistency regularization are not as strong compared to works from the computer vision domain.

These results demonstrate the benefits of TK-KNN's balanced sampling strategy and its use of the distances in the latent space.

### 5.1 Overconfidence in Pseudo-Labelling Regimes

A key observation we found throughout self-training was that the performance of existing pseudo-labelling methods tended to degrade as the number of cycles increased. An example of this is illustrated in Figure 3. Here we see that when a pre-defined threshold is used, the model tends to improve performance for the first few training cycles. After that point, the pseudo-labeling becomes heavily biased towards the easier classes. This causes the model to become overconfident in predictions for those classes and neglect more difficult classes. PL-Flex corrects this issue but converges much earlier in the learning process. TK-KNN achieves the best performance thanks to the slower balanced pseudo-labeling approach. This process helps the model learn clearer decision boundaries for all classes simultaneously and prevent overconfidence in the model in some classes.

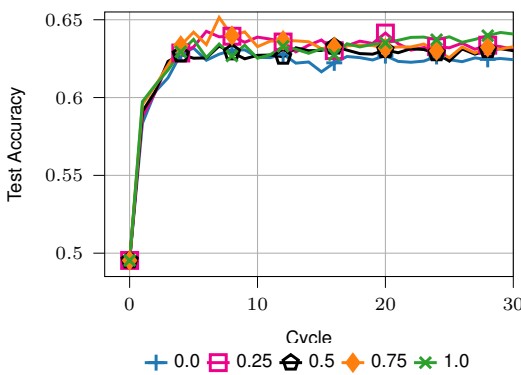

Figure 4: A comparison of TK-KNN on HWU64 with 1% labeled data as $\beta$ varies.

## 5.2 Ablation Study

Because TK-KNN is different from existing methods in two distinct ways: (1) top-k balanced sampling and (2) KNN ranking, we perform a set of ablation experiments to better understand how each of these affects performance. Specifically, we test TK-KNN under three scenarios, top-k sampling without balancing the classes, top-k sampling with balanced classes, and top-k KNN without balancing for classes. When we perform top-k sampling in an unbalanced manner, we ensure that the total data sampled is still equal to $k * C$, where $C$ is the number of classes. We report the results for these experiments in Table 3. We also conduct experiments on the addition of the different loss objectives detailed in Appendix C

The results from the ablation study demonstrate both the effectiveness of top-k sampling and KNN ranking. A comparison between our unbalanced sampling top-k sampling and balanced versions show a drastic difference in performance across all datasets. We highlight again that the performance difference is greatest in the lowest resource setting, with a **12.47%** increase in accuracy for CLINC150 in the 1% setting.

Results from the TK-KNN method with unbalanced sampling also show an improvement over unbalanced sampling alone. This increase in performance is smaller than the difference between unbalanced and balanced sampling but still highlights the benefits of leveraging the geometry for selective pseudo-labeling. We also present an ablation of the Top-k sampling methodology when passing correctly labeled examples in the Appendix A.

| Method | Percent Labeled | |
|---|---|---|
| | 1% | 2% |
| *CLINC150* | | |
| **Top-k U** | 38.37 ±1.08 | 55.0 ±1.44 |
| **Top-k B** | 51.36 ±2.1 | 64.99 ±0.64 |
| **Top-k KNN U** | 41.24 ±0.97 | 55.01 ±1.49 |
| **Top-k KNN B** | **53.73** ±1.72 | **65.87** ±1.18 |
| *BANKING77* | | |
| **Top-k U** | 41.56 ±4.73 | 54.78 ±3.51 |
| **Top-k B** | 50.45 ±4.53 | 63.19 ±1.78 |
| **Top-k KNN U** | 44.12 ±3.14 | 55.9 ±2.65 |
| **Top-k KNN B** | **54.16** ±4.56 | **62.71** ±2.30 |
| *HWU64* | | |
| **Top-k U** | 54.87 ±1.64 | 64.85 ±1.54 |
| **Top-k B** | 54.13 ±6.0 | 65.12 ±0.35 |
| **Top-k KNN U** | 57.86 ±2.25 | 69.33 ±0.96 |
| **Top-k KNN B** | **65.33** ±2.29 | **73.03** ±1.31 |

Table 3: Ablation study of top-k sampling. U stands for unbalanced sampling, where classes are not balanced. B is for balanced sampling, and classes are balanced with the top-k per class.

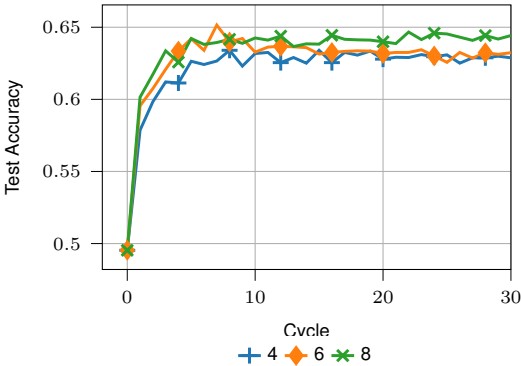

Figure 5: A comparison of TK-KNN on HWU64 with 1% labeled data as k varies.

## 5.3 Parameter Search

TK-KNN relies on two hyperparameters $k$ and $\beta$ that can affect performance based on how they are configured. We explore experiments to gauge their effect on learning by testing $k \in (4, 6, 8)$ and $\beta \in (0.0, 0.25, 0.50, 0.75, 1.00)$. When varying $k$ we hold $\beta$ at 0.75. For $\beta$ experiments we keep $k = 6$. When $\beta = 0.0$, this is equivalent to just top-k sampling based on Equation 3. Alternatively, when $\beta = 1.0$, this is equivalent to only using the KNN similarity for ranking. Results from our experiments are shown in Figures 4 for $\beta$ and 5 for $k$.

As we varied the $\beta$ parameter, we noticed that all configurations tended to have similar training patterns. After we trained the model for the first five cycles, the model tended to move in small jumps between subsequent cycles. From the illustration, we can see that no single method was always the best, but the model tended to perform worse when

$\beta = 0.0$, highlighting the benefits of including our KNN similarity for ranking. The model reached the best performance when $\beta = 0.75$, which occurs about a third of the way through the training process.

Comparison of values for $k$ show that TK-KNN is robust to adjustments in this hyperparameter. We notice slight performance benefits from selecting a higher $k$ of 6 and 8 in comparison to 4. When a higher value of $k$ is used the model will see an increase in performance earlier in the self-training process, as it has more examples to train from. This is only acheivable though when high quality correct samples are selected across the entire class distribution. If a $k$ value was selected that is too large, more bad examples will be included early in the training process and may result in poor model performance.

## 6 Conclusions

This paper introduces TK-KNN, a balanced distance-based pseudo-labeling approach for semi-supervised intent classification. TK-KNN deviates from previous pseudo-labeling methods as it does not rely on a threshold to select the samples. Instead, we show that a balanced approach that takes the model prediction and K-Nearest Neighbor similarity measure allows for more robust decision boundaries to be learned. Experiments on three popular intent classification datasets, CLINC150, Banking77, and Hwu64, demonstrate that our method improved performance in all scenarios.

## 7 Limitations

While our method shows noticeable improvements, it is not without limitations. Our method does not require searching for a good threshold but instead requires two different hyperparameters, $k$ and $\beta$, that must be found. We offer a reasonable method and findings for selecting both of these but others may want to search for other combinations depending on the dataset. A noticeable drawback from our self-training method is that more cycles of training will need to be done, especially if the value of $k$ is small. This requires much more GPU usage to converge to a good point. Further, we did not explore any heavily imbalanced datasets, so we are unaware of how TK-KNN would perform under those scenarios.

## 8 Ethics Statement

This work can be used to help improve current virtual assistant systems for many of the businesses that may be seeking to use one. As the goal of these systems is to understand a persons request, failures can lead to wrong actions being taken that potentially impact an individual.

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

---

**Algorithm 1** TK-KNN Sampling For a Cycle

---

**Input:** Data of $X = x_n, y_n : n \in (1, ..., N), U = u_m : m \in (1, ..., M), \beta$
  1: Predict pseudo-labels for all $U$
  2: Calculate cosine similarity via. Eq. (2) for all $U$ per class
  3: Calculate score via Eq.(3)
  4: Combine $X$ and top-k per class from $U$

---

## A Upper Bound Analysis

We further ran experiments to gauge the performance of top-k sampling when ground truth labels are fed to the model instead of predicted pseudo labels. This experiment gives us an indicator as to how performance should increase throughout the self-training process in an ideal pseudo-labeling scenario. We present the results of this in Figure. 6. As expected, the model tends to converge towards a fully supervised performance as the cycle increases and more data is (pseudo-)labeled. Another point of interest is that the method's upper bound can continue learning with proper labels, while TK-KNN method tends to converge earlier. The upper bound method also takes a significant increase in the first few cycles as well. This highlights a need to investigate methods for accurate pseudo label selection further, so that the model can continue to improve.

## B MixText Experiments

The MixText method requires a number of hyperparameters to be selected in order to acheive good performance. The method also relies on data augmentation of the original text. For our experiments we used the `Helsinki-NLP` models available on the huggingface repository [1] to perform back translation. Following the original paper we performed back translation into German (de) and Russian (ru). Our experiments used a labeled batch size of 4 and unlabeled batch size of 8, the same as the original paper. The Temperature parameter was set to 0.5.

---

[1] https://huggingface.co/Helsinki-NLP/opus-mt-en-de
https://huggingface.co/Helsinki-NLP/opus-mt-en-ru

## C Ablation Table

We present a detailed breakdown of our full ablation results in Table 4. These results include the performance when using only the cross-entropy loss, as well as various combinations of the supervised contrastive loss and differential entropy regularizer. The results in this table demonstrate that the inclusion of additional loss objectives improves performance. This is particularly evident when we add our KNN objective, as we observe an increase of approximately 1-4%. Although the addition of the differential entropy regularizer yields smaller performance improvements, it remains beneficial to our method overall.

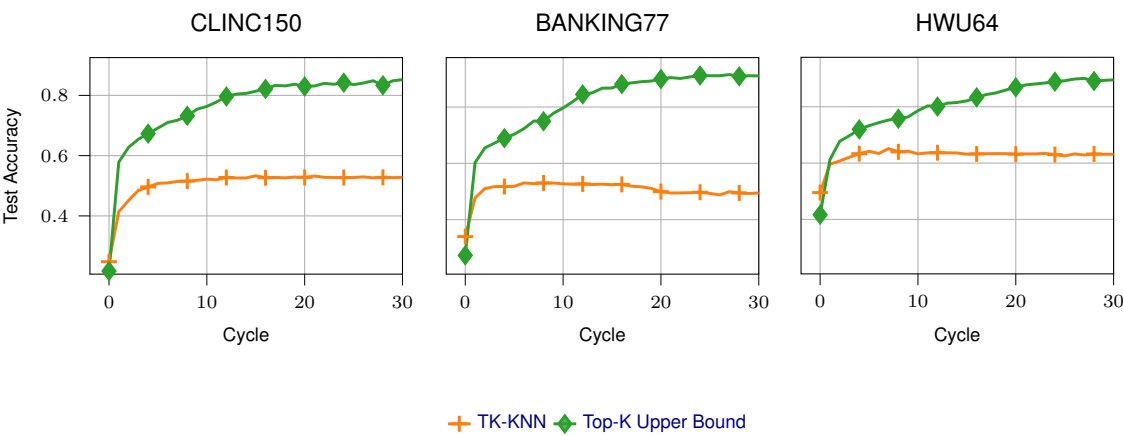

Figure 6: Ablation results for each dataset using 1% labeled data.

| Method | Percent Labeled | |
|---|---|---|
| | 1% | 2% |
| | CLINC150 | |
| **Top-k U CE** | 37.06 ±2.09 | 50.8 ±1.79 |
| **Top-k U CE CON** | 38.67 ±1.47 | 55.09 ±0.85 |
| **Top-k U CE CON DER** | 38.37 ±1.08 | 55.0 ±1.44 |
| **Top-k B CE** | 49.53 ±3.1 | 62.45 ±1.21 |
| **Top-k B CE CON** | 52.33 ±2.53 | 64.52 ±1.63 |
| **Top-k B CE CON DER** | 51.36 ±2.1 | 64.99 ±0.64 |
| **Top-k KNN U CE** | 40.58 ±2.54 | 53.45 ±1.05 |
| **Top-k KNN U CE CON** | 42.3 ±2.86 | 56.07 ±1.26 |
| **Top-k KNN U CE CON DER** | 41.24 ±0.97 | 55.01 ±1.49 |
| **Top-k KNN B CE** | 50.28 ±3.00 | 63.99 ±0.85 |
| **Top-k KNN B CE CON** | 53.23 ±3.17 | 65.37 ±1.17 |
| **Top-k KNN B CE CON DER** | **53.73** ±1.72 | **65.87** ±1.18 |
| | BANKING77 | |
| **Top-k U CE** | 36.19 ±2.93 | 51.69 ±3.54 |
| **Top-k U CE CON** | 42.16 ±4.61 | 54.77 ±2.48 |
| **Top-k U CE CON DER** | 41.56 ±4.73 | 54.78 ±3.51 |
| **Top-k B CE** | 44.35 ±3.82 | 57.19 ±3.23 |
| **Top-k B CE CON** | 50.48 ±3.75 | 62.34 ±1.57 |
| **Top-k B CE CON DER** | 50.45 ±4.53 | **63.19** ±1.78 |
| **Top-k KNN U CE** | 36.96 ±4.78 | 52.42 ±2.75 |
| **Top-k KNN U CE CON** | 44.86 ±3.53 | 56.49 ±2.68 |
| **Top-k KNN U CE CON DER** | 44.12 ±3.14 | 55.9 ±2.65 |
| **Top-k KNN B CE** | 49.16 ±3.02 | 59.81 ±1.69 |
| **Top-k KNN B CE CON** | 52.4 ±3.91 | 62.35 ±2.18 |
| **Top-k KNN B CE CON DER** | **54.16** ±4.56 | 62.71 ±2.30 |
| | HWU64 | |
| **Top-k U CE** | 51.45 ±3.8 | 62.79 ±1.99 |
| **Top-k U CE CON** | 54.8 ±2.1 | 66.58 ±1.17 |
| **Top-k U CE CON DER** | 54.87 ±1.64 | 64.85 ±1.54 |
| **Top-k B CE** | 61.69 ±3.15 | 70.41 ±0.66 |
| **Top-k B CE CON** | 63.05 ±2.6 | 72.55 ±1.41 |
| **Top-k B CE CON DER** | 54.13 ±6.0 | 65.12 ±0.35 |
| **Top-k KNN U CE** | 52.97 ±1.45 | 64.41 ±1.58 |
| **Top-k KNN U CE CON** | 59.26 ±2.99 | 69.98 ±1.44 |
| **Top-k KNN U CE CON DER** | 57.86 ±2.25 | 69.33 ±0.96 |
| **Top-k KNN B CE** | 62.43 ±2.78 | 71.04 ±0.93 |
| **Top-k KNN B CE CON** | 64.41 ±2.26 | 72.71 ±1.07 |
| **Top-k KNN B CE CON DER** | **65.33** ±2.29 | **73.03** ±1.31 |

Table 4: Ablation study of top-k sampling. U stands for unbalanced sampling, where classes are not balanced. B is for balanced sampling, and classes are balanced with the top-k per class. CE stands for cross-entropy loss, CON for contrastive loss, and DER differential entropy regularizer.