# OpenReview forum: "TK-KNN: A Balanced Distance-Based Pseudo Labeling Approach for Semi-Supervised Intent Classification"
_EMNLP/2023/Conference — EMNLP 2023 Findings_

### Official Review · Reviewer_67QQ · 2023-07-31

**Soundness:** 3

**Excitement:**

4: Strong: This paper deepens the understanding of some phenomenon or lowers the barriers to an existing research direction.

**Missing References:**

-

**Paper Topic And Main Contributions:**

The paper is a contribution to the problem of generating pseudo-labels of data in a supervised learning context when the amount of labeled data is small, a lot of unlabeled data is available or continuously generated and additional (manual, i.e. human) labeling should be avoided. It is well-known that a naive creation of pseudo-labels by using predictions from a model fit to a small training data set (usually with a threshold, fix or class dependent) leads into serious problems as the problem becomes biased towards labels which are easy to predict. This leads to unbalance in the classes, becomes increasingly worse if the procedure is iterated and leads finally to bad labels.
The paper proposes a new idea to avoid this problem.
Additionally, the number of classes in intent classification is rather high which is a challenge for the method. The new idea combines pseudo-labeling ideas (called top-K sampling) with distance based K nearest neighbor ideas from unsupervised learning (KNN).

**Questions For The Authors:**

1) You write on line 455ff:
"For TK-KNN, we set k = 6, 455, β = 0.75, and γ = 0.1 and report the results for these settings. An ablation study of these two hyperparameters is presented later."
As far as I see, only k and  β habe been used in the ablation study. Please comment on this.

2) 4400 GPU for the final experiment is a lot. Please comment for which steps you think you needed the most computing time.

**Reasons To Accept:**

The main strength of the paper is the proposition of a well-designed new method for learning from a small of number of labeled examples and by applying it to problems with a relatively large number of classes (e.g. one dataset has 150+1 classes).
The new approach is compared to other proposed methods including an lower bound and upper bound model with respect to the performance.
The approach seems to be not overly complex in the design - although very time consuming -, so that ideas may be easily translated to other supervised learning problems.
Confidence intervals are given for the results.
The results look strong compared to previous approaches (e.g. PL-Flex or MixText), especially if the percentage of already labeled data is small.

**Reasons To Reject:**

The ablation study seems to be incomplete (see the sectoin "Questions for the authors").
When the percentage of labeled data reaches 10%, the improvement to other methods is much less impressive.
As mentioned in the limitation section, the approach was not tested on highly imbalanced data.

**Reproducibility:**

3: Could reproduce the results with some difficulty. The settings of parameters are underspecified or subjectively determined; the training/evaluation data are not widely available.

**Reviewer Confidence:**

3: Pretty sure, but there's a chance I missed something. Although I have a good feel for this area in general, I did not carefully check the paper's details, e.g., the math, experimental design, or novelty.

**Typos Grammar Style And Presentation Improvements:**

-

---

> ### Author Rebuttal · Authors · 2023-08-28
>
> Thank you for your thoughtful feedback. We are encouraged to know that the paper proposes a well-designed new method that is not overly complex, and that the results look strong compared to previous approaches given the confidence intervals.
>
> We now respond to your comments in detail:
>
> > You write on line 455ff that use you 3 hyper-parameters for the ablation but I only see two used in the ablation study.
>
> Thanks for pointing out that we are missing an ablation study for the Gamma hyperparameter, we have included it in the paper. Our selection of 0.1 for Gamma in our work was based on previous literature cited in our paper [A] that uses this value. Below is an ablation table on the HWU dataset at 1% data for different values of gamma.  Our findings align with using the value 0.1 as a good default for Gamma and also show that the method is robust to variations in this hyper-parameter.
>
>
> | HWU Gamma | ACC ± CI    |
> |-----------|-------------|
> | 0         | 64.42 ± 2.08|
> | 0.05      | 64.74 ± 2.73|
> | 0.1       | 65.33 ± 2.29|
> | 0.15      | 64.44 ± 2.33|
> | 0.2       | 64.68 ± 2.24|
> | 1         | 63.1 ± 2.65 |
>
>
> [A] Sablayrolles, Alexandre, et al. "Spreading vectors for similarity search." arXiv preprint arXiv:1806.03198 (2018).
>
> > 4400 GPU for the final experiment is a lot. Please comment for which steps you think you needed the most computing time.
>
> The number of self-training cycles we selected likely contributed the most to this amount of GPU time, followed by the amount of ablation studies we conducted and the fact that we ran each experiment 5 times to get the mean and confidence interval.  After seeing our results and how the method tends to flatline after ~15 cycles we could have selected a lower number for the number of cycles, such as 20 or 25, and significantly reduced the total amount of compute needed.
>
> Fortunately, when we run an experiment on a single dataset and a single set of hyperparameters then the number of GPU hours would be around 3-4 hours, which is practical.

---

### Official Review · Reviewer_Rv94 · 2023-08-03

**Soundness:** 2

**Excitement:**

3: Ambivalent: It has merits (e.g., it reports state-of-the-art results, the idea is nice), but there are key weaknesses (e.g., it describes incremental work), and it can significantly benefit from another round of revision. However, I won't object to accepting it if my co-reviewers champion it.

**Missing References:**

The references discussed above. It is not comprehensive or even representative, but it marks some of the missing fields in the related work.

[1] Gera, Ariel, et al. "Zero-shot text classification with self-training." arXiv preprint arXiv:2210.17541 (2022).

[2] Wang, Ran, and Xinyu Dai. "Contrastive learning-enhanced nearest neighbor mechanism for multi-label text classification." Proceedings of the 60th Annual Meeting of the Association for Computational Linguistics (Volume 2: Short Papers). 2022.

[3] Chen, Junfan, et al. "Contrastnet: A contrastive learning framework for few-shot text classification." Proceedings of the AAAI Conference on Artificial Intelligence. Vol. 36. No. 10. 2022.


**Paper Topic And Main Contributions:**

This paper improves self-training of classifiers with the topk-kNN selection for the pseudo-labeled data points. Specifically, the authors propose  a scoring function that uses a weighted sum of predicted probability of pseudo-label of unlabeled data, and its cosine similarity to the nearest labeled instance in the predicted class. Then the topk scored instances in each class is used as labeled data for supervised training.

**Questions For The Authors:**

1. A major design choice of the scoring function is to use a distance of closest "predicted" instance to adjust the scoring. What is the motivation of that choice (why not using the prototype or label embedding)? How to justify a single instance is robust?

2. In the baselines, which ones employ self-training techniques?

3. [Optionally] How many iterations for self-training and what is the stop criteria?


**Reasons To Accept:**


1. Improving the stability of self-training is an important and hot topic in the NLP community.
2. The proposed methods show improvement in intent classification datasets, especially in low resource settings.


**Reasons To Reject:**

1. One of the concern is on the major contribution of the paper, as a balanced topk selection [1] is already a known trick to improve stability of self-training (including up-sampling and down-sampling label with different frequencies).
2. kNN-based classification [2],  prototypical network, and contrastive learning methods [3,4] are well established in the community. The related work should be discussed in order to claim novel contribution.

[1] Gera, Ariel, et al. "Zero-shot text classification with self-training." arXiv preprint arXiv:2210.17541 (2022).

[2] Wang, Ran, and Xinyu Dai. "Contrastive learning-enhanced nearest neighbor mechanism for multi-label text classification." Proceedings of the 60th Annual Meeting of the Association for Computational Linguistics (Volume 2: Short Papers). 2022.

[3] Chen, Junfan, et al. "Contrastnet: A contrastive learning framework for few-shot text classification." Proceedings of the AAAI Conference on Artificial Intelligence. Vol. 36. No. 10. 2022.




**Reproducibility:**

4: Could mostly reproduce the results, but there may be some variation because of sample variance or minor variations in their interpretation of the protocol or method.

**Reviewer Confidence:**

4: Quite sure. I tried to check the important points carefully. It's unlikely, though conceivable, that I missed something that should affect my ratings.

**Typos Grammar Style And Presentation Improvements:**

For equation (1), and (3), the notation is not very clear. Normally, p(y|x) in equation (1) means a probability distribution over label space, but in the context, it is the probability of the most confident prediction y (and y is the pseudo-label). For equation (3), Z_n is not clearly defined.

---

> ### Author Rebuttal · Authors · 2023-08-28
>
> Thank you for your thoughtful feedback. We are encouraged to know that the paper proposes methods that show improvement in intent classification datasets, especially in low resource settings, and that we address an important topic related to improving the stability of self-training which is a hot topic in the NLP community.
>
> We now respond to your comments in detail:
>
> > The references discussed above. It is not comprehensive or even representative, but it marks some of the missing fields in the related work.
>
> We would like to thank the reviewer for pointing out these recent relevant works. We will be adding all of these papers to our related work section and do a more thorough literature review. We have discussed these papers (and additional papers that are relevant to our task) in the related work section.
>
> > balanced topk selection [1] is already a known trick. kNN-based classification [2], prototypical network, and contrastive learning methods [3,4] are well established in the community
>
> In regards to [1], we address a different task than the zero-shot classification that the authors address. Further, our methods are different because we do not perform token masking, use NLI entailment methods, nor have to use strategies to select negative/positive examples.
>
> Further, we incorporate K-nearest neighbor in the embedding space and contrastive learning, thus our method is more finetuned towards the semi-supervised setup that we have. We also provided an ablation study in our work to show that our method achieves better results than if we used top-k alone (also shown in the table below).
>
>
>
> Paper [2] addresses a different task of multi-label classification rather than semi-supervised. Further, paper [2] uses contrastive learning for a very different objective than ours. While we use it to predict the right pseudo label using the embedding space, they use it to identify missing labels from similar examples. They also use this method for the full dataset whereas we use a variation of this method in the low data setting where we have to address stability issues related to that setup by including a balanced version of top-k example selection mechanism.
>
> [3] addresses a significantly different task of meta-learning in a few-shot setup, which also does not include a balanced way of performing self-training. However, both [2, 3] inspire the idea of using KNN + Contrastive learning, and we show in the table below that our method significantly outperforms this strategy.
>
> **Table 1**
> | Method | Top-k | KNN | Self-training | Contrastive Learning | CLINC Results 1% (Our Method) |
> |--------|-------|-----|---------------|----------------------|-------------------------------|
> | Inspired by [1]    | X     |     | X             |                      | 49.53 ±3.1                    |
> | Inspired by [2, 3]    |       | X   |  X $^\dagger$          | X                    | 42.3 ±2.86                    |
> | Ours   | X     | X   | X             | X                    | 53.73 ± 1.72                  |
>
>
> $^\dagger$ While the methods in [2,3] do not use self-training, we used it for our experiments to make them comparable.
>
>
> > A major design choice of the scoring function is to use a distance of closest “predicted” instance to adjust the scoring. What is the motivation of that choice? (why not use the prototype or the label embedding? How to justify a single instance is robust.
>
> Our justification that a single instance is robust relies on the fact that we only compare our pseudo labeled examples with the known instances that we have. Further, our justification for not picking the closest to the mean is that it will lead to lower variance than just choosing a single instance and could hurt diversity in the examples learned in subsequent self-training cycles.
>
> Our approach is shown to be robust as we ran experiments across 5 different random seeds for datasets to show that the performance does not generate a high degree of variance in results.
>
> We have also ran an experiment using a prototype showing that single instance outperforms the Prototype method (see Table below).
>
> | CLINC Experiments          | 1%            | 2%            | 5%            | 10%           |
> |--------------------------|---------------|---------------|---------------|---------------|
> | Prototype       | 51.9 ± 2.15   | 64.88 ± 0.86  | 74.29 ± 0.82  | 79.27 ± 1.09  |
> | Single Instance (Ours)   | 53.73 ± 1.72  | 65.87 ± 1.18  | 74.31 ± 0.96  | 79.45 ± 1.01  |
>
> > In the baselines, which ones employ self-training techniques?
>
> All baseline methods utilize self-training.
>
> > How many iterations for self-training and what is the stop criteria?
>
> As discussed in Section 4.3 of the paper, we used 30 cycles of self-training for our experiments and used early stopping on the validation set to determine when to stop the training procedure.
>
> > Typos Grammar Style And Presentation Improvements
>
> Thank you for suggesting grammar improvements. We have addressed the typos, improved the clarity of equation 1 by explaining what p(y|x) in the context mentioned in the paper and equation 3 by providing a clear definition for Z_n in the paper.

---

### Official Review · Reviewer_Z4is · 2023-08-06

**Soundness:** 3

**Excitement:**

3: Ambivalent: It has merits (e.g., it reports state-of-the-art results, the idea is nice), but there are key weaknesses (e.g., it describes incremental work), and it can significantly benefit from another round of revision. However, I won't object to accepting it if my co-reviewers champion it.

**Missing References:**

"it does not guarantee that discriminative features will be learned, ..." - needs citation


**Paper Topic And Main Contributions:**

In this paper, the authors focus on the problem of intent detection - an important first step in dialogue systems to understand the customer query intent and route the query accordingly. Typically posed as a multiclass classification problem, this requires access to large amounts of labelled data across the intent labels. This can be costly to get and, therefore, researchers have turned to semi-supervised techniques to alleviate this challenge. Specifically, one of the strategies is to generate high confidence pseudo labels for the unlabelled data using the classification model and use these to iteratively train the model. Here, the current research often resorts to using a fixed or flexible threshold across classes. The authors argue that this leads to biased sampling from over confident (easy) labels or samples with error labels due to low confidence that might adversely impact the model training.
They then propose an approach that they call TK-KNN that attempt to address these challenges. In their proposed approach, they dispense from using a threshold and instead sample top-k pseudo labelled instances from across intent labels irrespective of their prediction scores. While the model predictions can be noisy, the authors propose to address this by sampling top-k instances for a label that are also closer to their manually labelled counterparts.
The authors demonstrate the efficacy of their approach through evaluation on three recent intent classification datasets on varying sizes of labeled data.

**Questions For The Authors:**

Some questions above.
What is the "Top-k upper" setting? "select the correct pseudo label" - what does this mean? you never know whether the pseudo label is correct or not.
Is "k" fixed across cycles? Did you experiment with varying k? Perhaps this might help the issue of "more cycles of training will need to be done, especially if the value of k is small."
Equation 1.. what is u_m?

**Reasons To Accept:**

Data efficient deep learning is important from the perspective of cost and resource requirements as well as far reaching benefits to the environment. Further, conversational AI, the use case considered in this paper, has potential to save the high costs typically involved in customer support. This paper aims to address these and extends the state of art in these areas.
The authors propose sound ideas to address the bias related challenges in a pseudo labelling approaches. The evaluation supports their claims and consistently shows improved performance over multiple baselines.

**Reasons To Reject:**

The authors hypothesise that two points close in the latent space have the same output label. Am not sure if this always holds true in the context of the intent classification problem considered here. Data elements close in the latent space are 'semantically related' but not
necessarily in the same-label sense. They could have subsuming labels. “book_hotel” vs “change_hotel_booking” for instance. I would have liked to see more supporting evidence of this in the evaluation.

While the authors make a lot of interesting observations through their evaluation, the intuitions and reasoning are missing. For instance, GAN-BERT is unstable initially but improves with increase in the proportion of labeled data - Why?

**Reproducibility:**

4: Could mostly reproduce the results, but there may be some variation because of sample variance or minor variations in their interpretation of the protocol or method.

**Reviewer Confidence:**

4: Quite sure. I tried to check the important points carefully. It's unlikely, though conceivable, that I missed something that should affect my ratings.

**Typos Grammar Style And Presentation Improvements:**

"However, as well shall... " --> "However, as we shall ..."
"closely realted" --> "closely related"
"spread the representations our ..." --> "spread the representations out ..."
"ensures that our model with..." --> "ensures that our model will ..."

---

> ### Author Rebuttal · Authors · 2023-08-28
>
> Thank you for your thoughtful feedback. We are encouraged to know that the paper addresses an important problem in deep learning related to data efficiency, that we propose  "sound ideas to address the bias related challenges in a pseudo labelling approaches", and that the evaluation supports our claims and consistently show improved performance over multiple baselines.
>
> We now respond to your comments in detail:
>
> >  The authors hypothesise that two points close in the latent space have the same output label. Am not sure if this always holds true in the context of the intent classification problem considered here. Data elements close in the latent space are 'semantically related' but not necessarily in the same-label sense.
>
> It is true that points that are semantically close might not be close in the latent space for intent classification in the case of using pretrained LLMs without finetuning.
>
> However,  we are finetuning these LLMs with the contrastive loss function to ensure that the label representations for the same class are pulled together and different ones will be forced apart, encouraging points with the same output label  to be close in the latent space.
>
> > For instance, GAN-BERT is unstable initially but improves with increase in the proportion of labeled data - Why?
>
> Previous studies have shown that training GANs with too little data can cause overfitting in the discriminator causing training to diverge [A, B], which aligns with our GAN-BERT experiments.
>
> - [A] Arjovsky, Martin, and Léon Bottou. "Towards principled methods for training generative adversarial networks." arXiv preprint arXiv:1701.04862 (2017).
>
> - [B] Salimans, Tim, et al. "Improved techniques for training gans." Advances in neural information processing systems 29 (2016).
>
> > What is the top-k upper setting?
>
> top-k upper is the upper bound performance that would be achieved if all selected examples have the correct pseudo label. It gives us the highest potential improvement we could make in our setting and identify how close our proposed method is to the upper bound.
>
>
> > “Select the correct pseudo label” -What does this mean? You never know whether the pseudo label is correct or not.
>
> This is an experiment we conducted to see the best performance possible if you give the model the ground truth result at each phase of pseudo labeling (it can also be considered as the oracle performance). We have clarified what we mean by “correct pseudo label” in Section 4.2.
>
> > Is “k” fixed across cycles?
>
> The value of k is fixed across all cycles.
>
> > Did you experiment with varying “k”?
>
> We present in the paper an ablation study that we conducted with varying the value of k for our experiments (please see Figure 5). Our findings showed that our method is generally robust to the value of k.
>
> > Equation 1 what is u_m?
>
> U_m is a typo and is meant to be x_m which refers to an unlabeled example that we are pseudo labeling. We have fixed the typo in the paper
>
> > Typos Grammar Style And Presentation Improvements
>
> Thank you for pointing out fixes to several typos we had. We have addressed them in the paper.
>
> > "it does not guarantee that discriminative features will be learned, ..." - needs citation
>
> We thank you for pointing this out and have added the following citation [C] to our paper in this location.
>
> [C] Elsayed, Gamaleldin, et al. "Large margin deep networks for classification." Advances in neural information processing systems 31 (2018).

---

### Meta-Review · Area_Chair_LmBg · 2023-09-05

**Recommendation:** 4
**Best Paper Recommendation:** No

**Metareview:**

The paper proposes a new method, TK-KNN, for acquiring semi-supervised data for the task of intent detection. The reviewers agree that the problem itself is quite important, that the proposed method is a meritorious contribution towards solving the problem of acquiring labeled data for intent classification, and that the experimental results appear to be sound. Each reviewer has some different individual concerns; R1 about the theoretical motivation behind the method, which the authors try to clarify; R2 about novelty, as top-k KNN + contrastive learning is already a well established technique on other tasks; and R3 about the experimental setup, particularly that more ablations could be useful. Of these, the most salient concern is about novelty: the authors note that the particular combination of top-k KNN + contrastive learning has not been studied for semi-supervised learning on the task of intent classification, and is therefore still a good contribution, while R2 holds that these are well established techniques and it already makes sense that they would perform well in this setting. The study appears to be overall well executed, and though the method itself may not be particularly exciting from a novelty perspective, its application to the task of intent classification with extreme number of labels is still a potentially useful contribution.

**Meta-Review:**

The paper proposes a new method, TK-KNN, for acquiring semi-supervised data for the task of intent detection. The reviewers agree that the problem itself is quite important, that the proposed method is a meritorious contribution towards solving the problem of acquiring labeled data for intent classification, and that the experimental results appear to be sound. Their main concerns are that the results may not be so surprising as similar methods have been successful in other domains, and some further analysis explaining why their proposed method works would benefit the paper. Additionally, R3 points out that more ablations would be useful. The authors clarify some of the differences between their proposed method and previous work, particularly that the application of these methods in combination for the problem of multi-label classification is novel w.r.t. the literature.

---

### Decision · Program_Chairs · 2023-10-07

**Decision:**

Accept-Findings

**Comment:**

The paper proposes a new method, TK-KNN, for acquiring semi-supervised data for the task of intent detection. The reviewers agree that the problem itself is quite important, that the proposed method is a meritorious contribution towards solving the problem of acquiring labeled data for intent classification, and that the experimental results appear to be sound. Each reviewer has some different individual concerns; R1 about the theoretical motivation behind the method, which the authors try to clarify; R2 about novelty, as top-k KNN + contrastive learning is already a well established technique on other tasks; and R3 about the experimental setup, particularly that more ablations could be useful. Of these, the most salient concern is about novelty: the authors note that the particular combination of top-k KNN + contrastive learning has not been studied for semi-supervised learning on the task of intent classification, and is therefore still a good contribution, while R2 holds that these are well established techniques and it already makes sense that they would perform well in this setting. The study appears to be overall well executed, and though the method itself may not be particularly exciting from a novelty perspective, its application to the task of intent classification with extreme number of labels is still a potentially useful contribution.